# Observation of boundary induced chiral anomaly bulk states and their transport properties

Mudi Wang[1], Qiyun Ma[2], Shan Liu[2], Ruo-Yang Zhang [1], Lei Zhang [3,4], Manzhu Ke [2], Zhengyou Liu [2,5] ✉ & C. T. Chan [1] ✉

The most useful property of topological materials is perhaps the robust transport of topological edge modes, whose existence depends on bulk topological invariants. This means that we need to make volumetric changes to many atoms in the bulk to control the transport properties of the edges in a sample. We suggest here that we can do the reverse in some cases: the properties of the edge can be used to induce chiral transport phenomena in some bulk modes. Specifically, we show that a topologically trivial 2D hexagonal phononic crystal slab (waveguide) bounded by hard-wall boundaries guarantees the existence of bulk modes with chiral anomaly inside a pseudogap due to finite size effect. We experimentally observed robust valley-selected transport, complete valley state conversion, and valley focusing of the chiral anomaly bulk states (CABSs) in such phononic crystal waveguides. The same concept also applies to electromagnetics.

The discrete valley degree of freedom, which is a simple yet elegant way to enable topologically protected transport, is attracting much attention in both electronics[1–9] and classical wave fields[10–19] due to its simplicity. Similar to the spin in spintronics, the valley index is an information carrier. Moreover, chiral valley transport[8–18] is robust against weak disorder, as the intervalley scattering is usually small due to the large separation of the valleys in momentum space. In the valley transport realized to date, the most common approach is to create topological boundary states[8–18] at the interface of two crystals carrying different valley Chern numbers according to the bulk-edge correspondence[2–21] and the valley Chern numbers are created by lowering/breaking the symmetry of the bulk crystal.

Here, we propose a mechanism for providing robust transport using chiral anomaly bulk states (CABSs), and the effect is phenomenologically the same as valley transport. The CABSs require only boundary modifications of a simple non-topological crystal, which apply to both acoustics and electromagnetics. Such simplicities are helpful for a wide range of practical applications. In addition,

boundary-condition controlled valley-selected transmission can be easily implemented, as we will show below. Optical frequency resonance structures based on CABSs can behave like valley Hall topological laser systems if the photonic crystal has gain, except that the amplified light comes from chiral bulk modes rather than boundary modes.

## Results

### Construction of boundary-induced CABSs
We consider a hexagonal PC slab (lattice constant $a$ is 12.0 mm) composed of circular rods (7.0 mm diameter, 10 mm thickness) with a rigid boundary (on the $xz$ plane) imposed front and back, which are axes perpendicular to the top and bottom plates ($xy$ plane), as shown in Fig. 1a. Such a configuration is effectively a waveguide. The bulk band diagram of the periodic 2D PC is shown in Fig. 1b. The mirror-$y$ ($M_y$) symmetry of the PC's unit cell about the middle line (dashed line) ensures that the bands can be classified as $M_y$ even or odd along Γ-K, as shown in the inset of Fig. 1b. For the even mode (red band), the normal

[1]Department of Physics, The Hong Kong University of Science and Technology, Hong Kong, China. [2]Key Laboratory of Artificial Micro- and Nanostructures of Ministry of Education and School of Physics and Technology, Wuhan University, Wuhan, China. [3]State Key Laboratory of Quantum Optics and Quantum Optics Devices, Institute of Laser Spectroscopy, Shanxi University, Taiyuan 030006, China. [4]Collaborative Innovation Center of Extreme Optics, Shanxi University, Taiyuan 030006, China. [5]Institute for Advanced Studies, Wuhan University, Wuhan, China. ✉e-mail: zyliu@whu.edu.cn; phchan@ust.hk

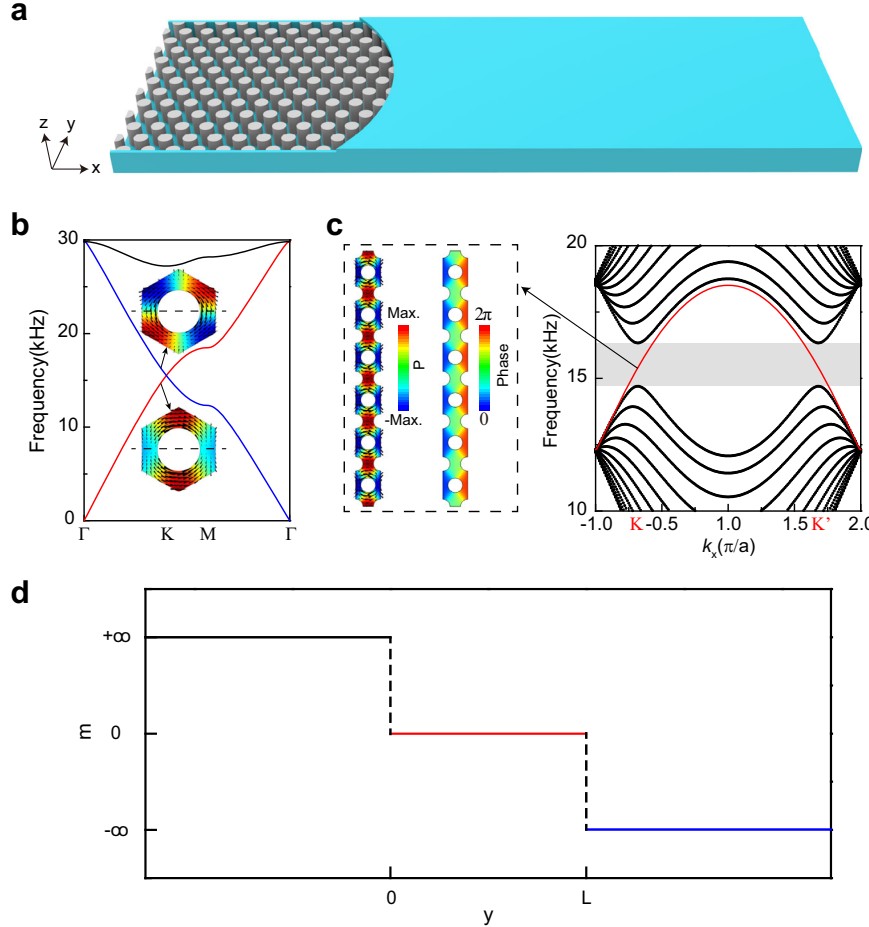

**Fig. 1 | Boundary condition induced CABSs. a** Schematic of the PC with a hard boundary. **b** Bulk band diagram. Inset images: the corresponding pressure (colors) and energy flux fields (arrows) of the eigenstates in the unit cell. The red (blue) band is even (odd) with respect to the mirror plane marked by the dotted line. **c** Band dispersion of the PC with a hard boundary, where the periodic boundary condition is in the x-direction and its width is 12 layers (each layer is $\frac{\sqrt{3}}{2}a$ wide) along the y-direction. Left panel: pressure (colors) and energy flux (arrows) fields and the pressure phase (colors) field of the CABS. The pseudogap (shadow region) is caused by the finite width of the PC waveguide. **d** The distribution of Dirac mass $m$ along y-direction in the continuum model (see text).

velocity fields of the eigenstates are zero at the mirror plane (black dashed line), while for the odd mode (blue band), the velocity field of the eigenstates is the maximum at the mirror plane. These modes cross at the K point.

When the front and back rigid boundaries are imposed by terminating the PC along the middle lines of the unit cells in the outermost layers of the sample, the normal velocity component must be zero, so only the even bulk mode with zero normal velocity component at the mirror plane is compatible with the boundary condition, while the odd modes with a non-zero normal velocity component in the mirror plane is incompatible with the hard-wall boundary condition and hence forbidden. Hence, a boundary-condition imposed chiral anomaly can emerge in the PC, as shown by the red band in the projected band structure of the PC waveguide in Fig. 1c. These chiral states are bulk modes, as they are spatially extended and independent of the width of the PC waveguide. They are K or K' valley-locked in the pseudoband gap (the shaded area in the right panel of Fig. 1c, which is caused by the finite size effect). The left panel of Fig. 1c shows the pressure (colors) and energy flux (arrows) fields and the pressure phase (colors) field of the CABSs.

The fact that boundary conditions at the edge can select the parity of allowed bulk mode can be understood heuristically using a simple model in the continuum limit. Suppose that the thickness of the waveguide is $L$, two hard boundaries are located at $y = 0$ and $L$. We can

consider a Dirac-like Hamiltonian

$$H(y) = \sigma_x k_x v_D - i\partial_y \sigma_y v_D + m(y)\sigma_z \qquad (1)$$

near the K valley, with a step mass term

$$m(y) = \begin{cases} m_1 & y<0 \\ 0 & 0 \leq y \leq L \\ m_2 & y>L \end{cases}. \qquad (2)$$

In the domain of $0 \leq y \leq L$ side, the Hamiltonian has a bulk mirror-y symmetry:

$$\sigma_x H(y) \sigma_x = H(-y). \qquad (3)$$

The Eigen solutions at $k_x = 0, \omega = 0$ are

$$\psi = \begin{cases} e^{|m_1|y}\begin{pmatrix} 1 \\ \mathrm{sgn}(m_1) \end{pmatrix} & y \leq 0 \\ \begin{pmatrix} 1 \\ \pm 1 \end{pmatrix} & 0 \leq y \leq L \\ e^{-|m_2|(y-L)}\begin{pmatrix} 1 \\ -\mathrm{sgn}(m_2) \end{pmatrix} & y \geq L \end{cases}. \qquad (4)$$

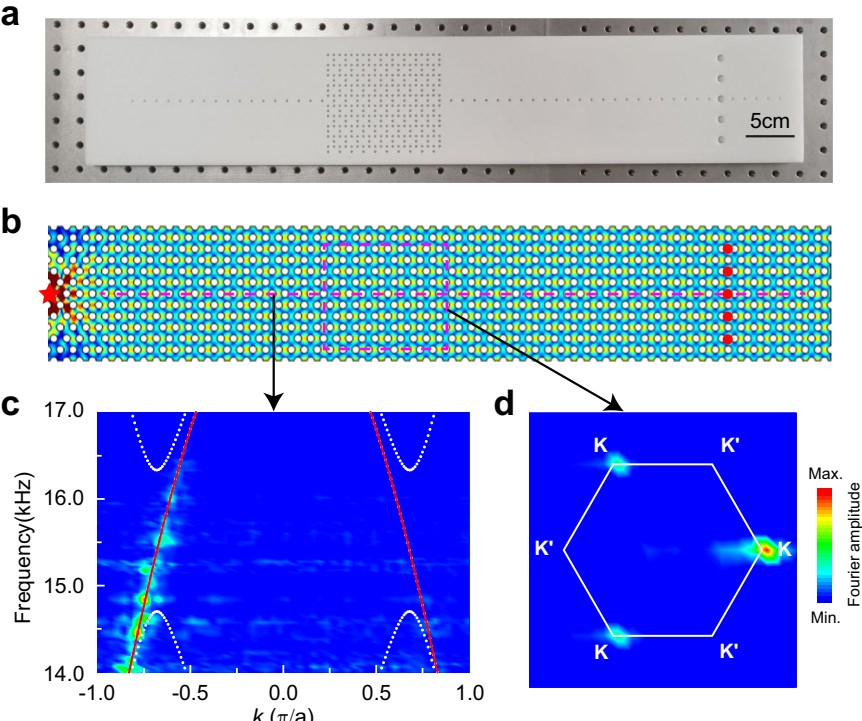

**Fig. 2 | Experimental measurement of the CABSs. a** Photo of the sample. **b** Simulated acoustic pressure field distributions at 15.5 kHz for a point source (red star) on the left of the straight PC. **c** Experimentally measured dispersions (colors) compared to the simulated dispersions (white dots and red lines). **d** Experimental Fourier spectra (indicated by a color map) at a frequency of 15.5 kHz, as obtained by a Fourier transformation of the measured field distribution inside the domain delineated by the magenta dashed lines in **b**.

We note that the mass terms (either positive or negative) give a gap and hence an exponentially decaying solution. The continuity of the wave function at $y = 0$ and $L$ requires that this equation have only two solutions: (1) $m_1 > 0$, $m_2 < 0$, and $\psi = \binom{1}{1}$ (bulk even mode with respect to mirror-$y$ operator $\sigma_x$) when $0 \leq y \leq L$; (2) $m_1 < 0$, $m_2 > 0$, and $\psi = \binom{1}{-1}$ (bulk odd mode) when $0 \leq y \leq L$. This means that when the two $m(y) \neq 0$ domains ($y \leq 0$ and $y \geq L$) have opposite mass terms, the sign of the Dirac masses $\text{sgn}(m_1) = -\text{sgn}(m_2)$ can select only one of the bulk modes with a certain mirror-$y$ parity (a Jackiw–Rebbi zero mode within this framework)[22] to be confined in the middle domain. In the limit $|m_1| = |m_2| = m_0 \rightarrow +\infty$, the waves cannot penetrate the two outside domains. Since the real boundary of the acoustic waveguide is rigid, where the normal velocity component must be zero, only the mirror-$y$ even bulk mode at the Dirac point is compatible with the rigid boundary condition. Compared with the above effective Dirac Hamiltonian model, we found that the Dirac step mass with $m_1 = -m_2 \rightarrow +\infty$ rather than the other case with $m_1 = -m_2 \rightarrow -\infty$ can be mapped to the rigid acoustic boundary of the waveguide, since only the former one can support a mirror-$y$ even bulk mode $\psi = \binom{1}{1}$. This means the step Dirac mass with $m_1 = -m_2 \rightarrow +\infty$ is the sole solution, as shown in Fig. 1d, which endows the CABSs (Fig. 1c). The sign of the mass term depends, of course, on the selection of one particular valley (the K valley) and the mass terms will switch signs if we consider the K′ valley instead. It is known that tuning the boundary potential of graphene can result in chiral states[23–29]. Here, we show that simply terminating the PC along the middle lines of the unit cells in the outermost layers of the sample with a hard boundary can guarantee the existence of the chiral states by symmetry compatibility selection (See Figs. S1, S2 and the corresponding descriptions for the other boundaries).

We note that the projected band structure in Fig. 1c is essentially the same as that of a finite strip of topological valley crystal[2–18], and as such, we expect similar transport properties, as will be shown later. However, the mechanism here is essentially different from a valley crystal. The usual topological valley crystal achieves chiral boundary modes by employing a carefully designed bulk material characterized by a bulk topological invariant (valley Chern number). Here, the PC with mirror symmetry achieves CABSs by employing a particular edge configuration: the mirror symmetry of the PC ensures that there exists an even mode and an odd mode, while the rigid boundary at the mirror plane of the unit cell allows only one mode (either even mode or odd mode) to propagate.

**Experimental observation of the CABSs**

Experimental samples, as shown in Fig. 2a, are fabricated with 3D printing technology using resin, which can be regarded as an acoustically rigid material. We drill some holes on the top cover of the sample, through which sound pressure and phase can be measured. The acoustic field distribution at 15.5 kHz is shown in Fig. 2b. For each excitation frequency, we measure the pressure field along the horizontal magenta dashed line shown in Fig. 2b. The corresponding Fourier transform gives the dispersion along $k_x$. The color map of Fig. 2c shows the measured intensity distribution in momentum space in the frequency window between 14.0 and 17.0 kHz, indicating the dispersion relation of the CABSs around the K valley (note the absence of intensity at the K′ valley). This result agrees well with the theoretical dispersion plotted with the red lines. Fixing the excitation frequency at 15.5 kHz, the pressure field distribution (containing both the amplitude and phase information) in the rectangle delineated by the magenta dashed lines is measured, and the Fourier spectrum is shown in Fig. 2d. We note that only states near the K valley are detected in the experiment, indicating the excitation of CABSs locked at the K-valleys. This is consistent with the

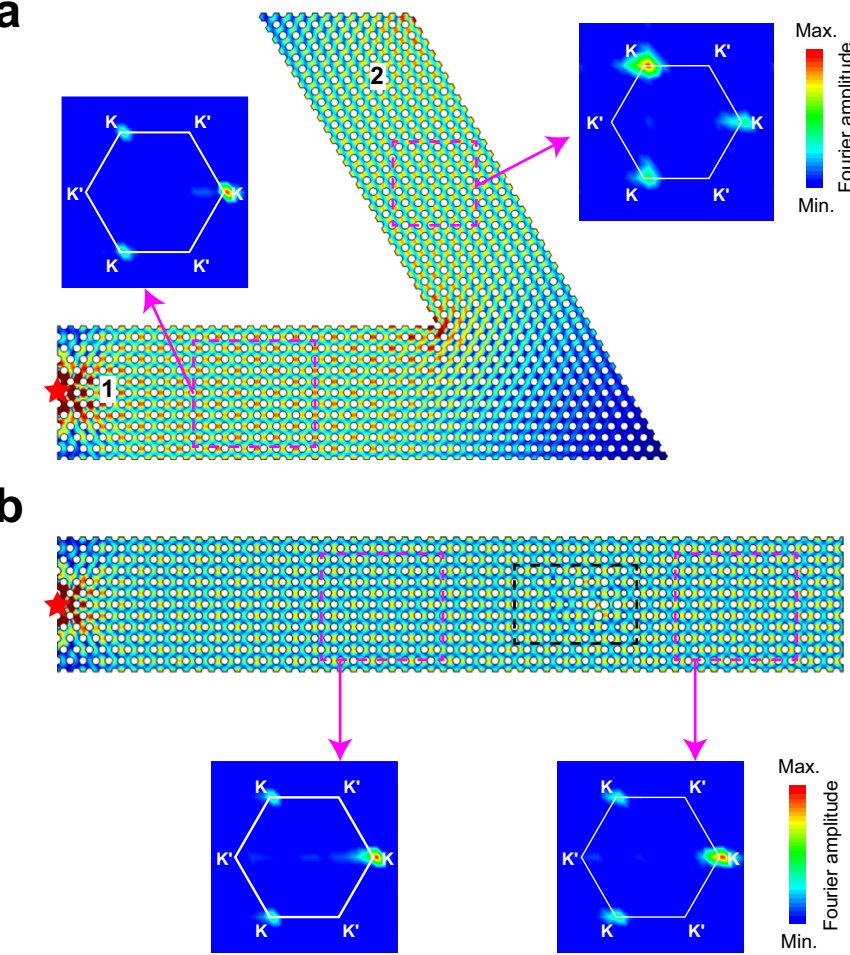

**Fig. 3 | Robustness of the CABSs against defects. a, b** Simulated acoustic pressure field distributions at 15.5 kHz for a point source (red star) on the left of the 120-degree-bend PC and the disordered PC (the cylinder diameters of the 32 units are randomly distributed between 5–9 mm in the black rectangle). Inset picture: the corresponding experimental Fourier spectra of the acoustic field in the magenta dashed rectangle in **a, b**.

dispersion relation of the CABSs shown in Fig. 1c, indicating a positive group velocity at K. The experimental results agree well with the theory.

### Properties of the CABSs

Due to the valley-locking property, CABS transport should have immunity against sharp bends or disorder at the same level of protection as valley Hall type topological transport. To verify this, PC waveguides with a 120-degree bend and some disorder in the interior (cylinder diameter randomly distributed between 5–9 mm) are fabricated, as shown in Fig. 3a, b, respectively. The acoustic point source is placed on the left of the two PCs, and the distribution of the acoustic field (15.5 kHz) remains extended, which means that the sharp bend or weak disorder does not compromise the transmission of the sound wave. The observed robustness of transport can be understood as follows. For Fig. 3a, the PCs in domains 1 and 2 have the same projected band structure, and they all permit the sound wave to pass through the K valley from the left. For Fig. 3b, despite the introduction of structural disorder inside the PC (as specified in the figure caption), the sound wave can pass through the PC without noticeable transport deterioration. This occurs because the CABSs are valley-locked and intervalley scattering from the K valley to the K' valley is weak, as the valleys have a large separation in momentum. In the experiment, we scan the acoustic field inside the domains enclosed by the magenta rectangles in Fig. 3a, b, and their Fourier spectra (the experimental results of some other frequencies are

shown in Fig. S3, and the theoretical transmission contrast is shown in Fig. S4) all show that for the frequencies in the CABS region, only the states near the K valley are excited, while the K' valley states are strongly suppressed.

In the previous works[8–18], robust transport enabled by topological boundary states was adjusted by controlling bulk topological invariants. If we want to change the existence and the property of the edge modes, the bulk should be changed. In a way, we modify the properties of many atoms to change the edge property, which is of measure zero compared to the bulk. However, the transmission of CABSs here can be controlled conveniently by modifying the boundary. This is simpler than other methods in the sense that we are modifying a few atoms, which then change the transport of the bulk in a certain frequency range. As shown in Fig. 4a, the sound wave cannot pass through the 60-degree-bend PC without reflection, as the projected band in domain 1 along the x-direction and domain 3 along the x' direction (shown in Fig. 4d, e) have different valley polarizations for the rightward propagative wave group velocity. For Fig. 4b, the width of PC is reduced by $\frac{\sqrt{3}}{2}a$ in domain 4 compared with that in domain 1, and the projected band is changed, as shown in Fig. 4f. Both domain 1 and domain 4 allow CABSs with K valley states to exist and hence, the sound wave can pass through the 60-degree-bend PC. This shows that the group velocity of CABSs in the valleys can be flipped by changing only the width of the PC (band dispersion regulation by changing its width is shown in Fig. S2), thereby allowing transport control by adjusting the edge morphology. As shown in

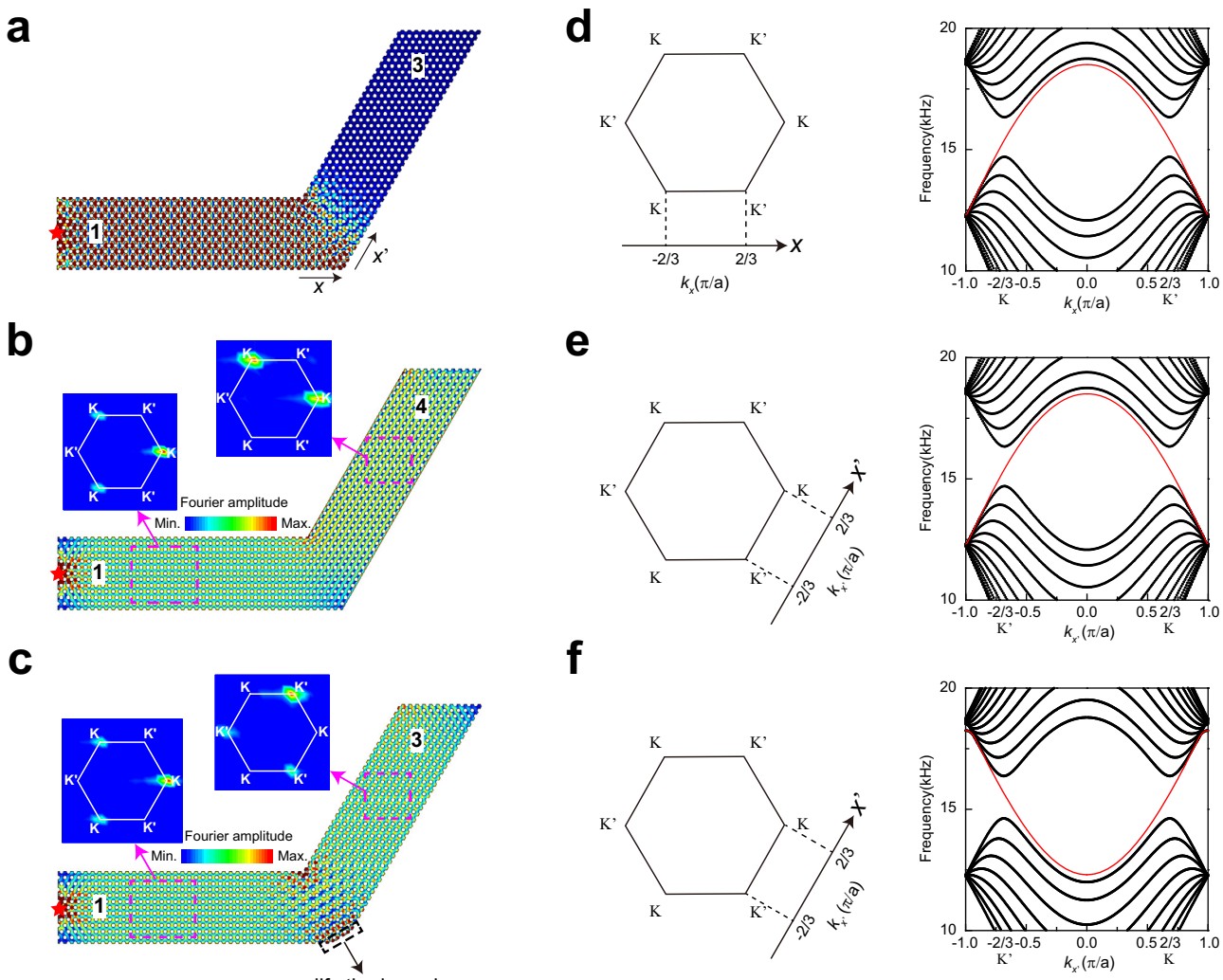

**Fig. 4 | Control the transmission of CABSs by adjusting the boundary modifications. a**–**c** Simulated acoustic pressure field distributions at 15.5 kHz for a point source in three different configurations of the 60-degree-bend PCs. The PC in **a** is composed of two waveguides with the same band dispersion (with periodic boundary conditions in the *x*- and *x'*-directions). The PC in **b** is composed of two waveguides with inverted K and K' valleys in the band dispersion. The width of the waveguide in domain 4 is smaller than that of domain 1 by one layer. The PC in **c** is composed of two waveguides with the same band dispersion, featuring some

boundary modifications at the coupling position (compared with **a**, a rigid boundary is added at 60 degrees to the *x* axis at the position 2*a* away from the bottom right corner along the -*x* axis). The modification of **b**, **c** is shown in Fig. S5. The inset pictures in **b**, **c** are the experimental Fourier spectra of the acoustic field in the corresponding magenta dashed rectangles. **d**–**f** Fourier spectra in momentum space for domains 1, 3, and 4. The right panels display the corresponding projected band structures along the *x*, *x'* and *x'* directions.

Fig. 4b, this is verified experimentally that the wave (15.5 kHz) excited by a source on the left can transmit through the sharp bend and only K valley states are excited in domains 1 and 4 and the K' valley states are deeply suppressed.

The transport of bent CABSs can also be controlled by the characteristics of the boundary of the waveguide at the joint. For example, we modify the boundary of the PC in Fig. 4a to the configuration shown in Fig. 4c. The only difference is the morphology of the boundary at the joint. With this very small change (Fig. S6 shows the influence of boundary morphologies), the sound wave (15.5 kHz) can pass through this new configuration from domain 1 to domain 3, meaning that the valley polarity completely changes from the K valley to the K' valley by edge modification at the joint (See more discussions in Supplementary Information). In photonic crystal waveguides, it is known that good coupling can be achieved by designing coupling regions[30–33] by facilitating impedance matching or by providing resonance coupling. However, the mechanism is rather different, as these conventional waveguide states carry neither chiral anomaly nor valley information.

In the chiral anomaly approach, the mechanism works for the entire frequency range inside the pseudogap, and the transport remains a single mode that is independent of the width of the waveguide. This phenomenon is confirmed by the experimental Fourier spectra in the area enclosed by the magenta dashed rectangles in Fig. 4c, which show that the CABSs are K-locked in domain 1 and K'-locked in domain 3, and the reflection is greatly suppressed during transmission. The complete valley conversion shown here has potential applications in valley information transforms[1–19].

The CABS also allow for the valley-locked focusing of waves by designing a suitable boundary configuration. Figure 5a shows the transport of the CABSs excited by a point source on the left through a slopped domain, where the layer number drops sharply from 12 to 2. The Fourier spectrum of the field in the area enclosed by the magenta dashed rectangle shows that only the K valley states are excited. This means that in this area, forward-moving CABSs (locked to K) channel the energy into the narrow guide on the right, with no noticeable reflection to the CABSs locked to K'. Moreover, we measure the

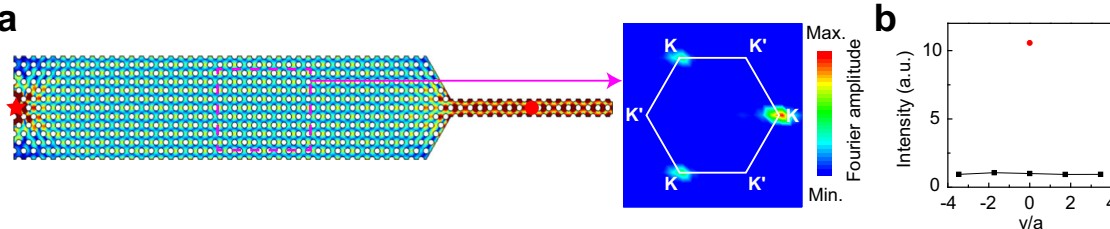

**Fig. 5 | Valley-locked focusing based on the CABSs. a** Simulated acoustic pressure field distributions at 15.5 kHz for a point source (red star) on the left of the PC, demonstrating the focusing effect. Right panel: the experimental Fourier spectrum of the acoustic field in the magenta dashed rectangle. **b** The black squares and red circle represent the experimentally measured intensity profiles along the red dotted lines in Figs. 2b, 5a, respectively.

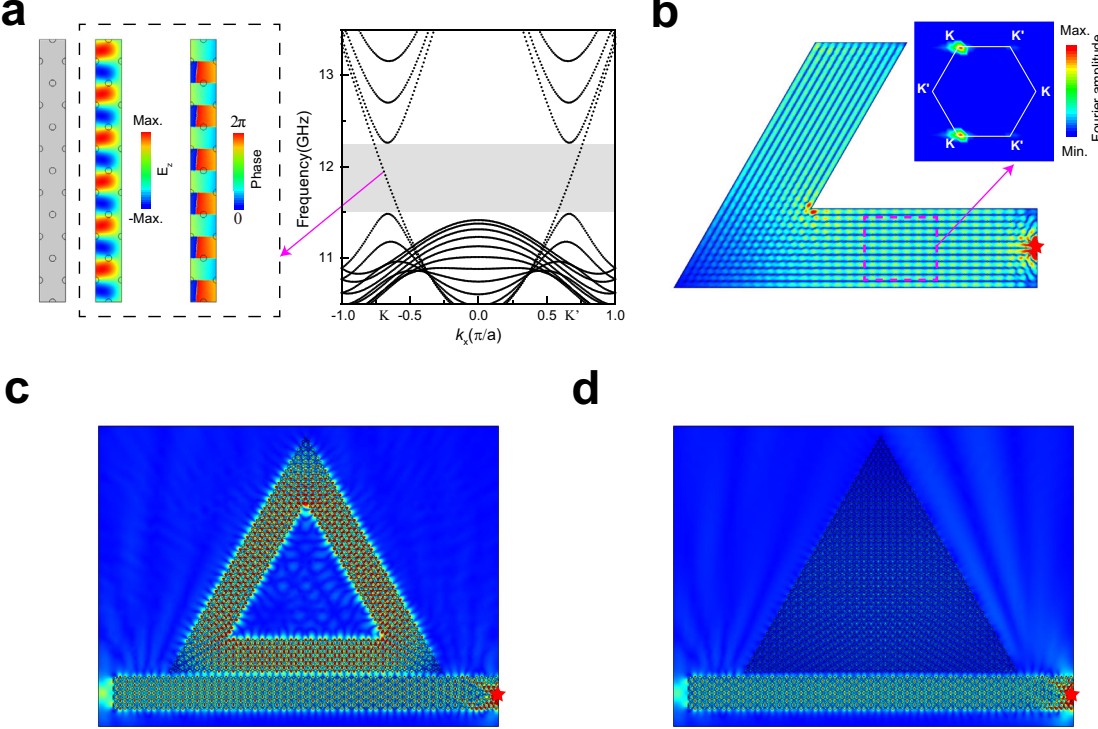

**Fig. 6 | Triangular waveguide resonator based on CABSs. a** Photonic crystal waveguide (microwave frequencies) bounded by perfect electrical conductor boundaries above and below and periodic boundaries on the left and right. Right panel: band dispersion of the photonic crystal waveguide and the corresponding electric field (colors) and the phase (colors) field of a CABS in the pseudogap (shadowed region). **b** Simulated electric field distributions at 11.9 GHz excited by a point source (red star) on the right of the 120-degree-bend waveguide. Right panels: the corresponding Fourier spectra of the electric fields in the magenta dashed rectangle. **c** Triangular waveguide resonator (the layer number is 8 and the permittivity of the dielectric is $\varepsilon = 12 + 0.002*i$) coupled with an eight-layer waveguide which is excited by a point source on the right at 482 THz (light wave frequencies). Note the uniformity of the resonance mode. **d** The resonator is changed to a solid triangle with the same bulk system parameters. Note that the resonance disappeared.

pressures at the red points in Figs. 2b, 5a, and the results show a significant intensity enhancement in the narrow channel. This valley-focusing effect may be useful in field enhancement or energy harvesting.

### Triangular waveguide resonator based on the CABSs

The concept of using boundary conditions to induce chiral anomaly bulk states is not limited to acoustic waves and it also works for photonic crystals. In Fig. 6a, we consider a dielectric photonic crystal waveguide operating at microwave frequencies. The waveguide is a 12-column 2D photonic crystal composed of a hexagonal lattice of dielectric cylinders ($\varepsilon = 12$) in air (the unit cell is shown in Fig. S7) with metallic boundaries on the top/bottom of the waveguide. While for light wave frequencies (see Fig. S8), we use a dielectric strip ($\varepsilon = 12$) which has an eight-column hexagonal array of air holes in the air. As the

metallic boundary condition is inconvenient to implement in practice in optical frequencies, we use an open boundary condition instead, i.e., the waveguide strip forms an interface with air $\varepsilon = 1$. From the band dispersion diagrams in Fig. 6a for microwave frequencies and Fig. S8b for optical frequencies, CABSs locked to the K (K′) valleys are found to exist in the pseudogap formed because of the finite width effect. As the transport is determined by the dispersion, we expect the same valley-locked transport characteristics in EM waves as we have demonstrated experimentally for acoustics. Such phenomena are indeed demonstrated numerically, as shown in Fig. 6b (for microwave frequency) and Fig. S8c (for light wave frequency), where we see high transmission through sharp bends.

Taking advantage of the CABSs, we design a triangular resonator that operates at optical frequencies as shown in Fig. 6c (the unit cell has the same parameters as those specified Fig. S8, except that we

added a very small gain so that $\varepsilon = 12 + 0.002*i$). The triangular resonator can be excited by a straight 8-column waveguide. The widths of the waveguide and the triangular resonator are all $4\sqrt{3}a$, and the spacing between the waveguide and the resonator is $\frac{\sqrt{3}}{2}a$, as shown in Fig. S9, and they have the same CABSs. Due to the valley-locked transport, there is no back-reflection at the sharp bends. As a consequence, the mode is uniform, and there is no standing wave node/anti-node intensity variation along the wave-propagation direction. In addition, and in contrast to valley Hall resonators and other topological laser systems[34–37], the mode is uniform even across the cross-section of the waveguide because it is a chiral bulk mode and not an edge mode. Such uniformity of intensity can avoid spatial hole burning and optimize the utilization of the gain medium during light amplification. In Fig. 6d, we show for comparison when the resonator is a solid triangle (i.e., no empty space inside). The configuration in Fig. 6d has more bulk modes than that in Fig. 6c, but the resonance effect has become significantly weaker, and the Q factor is about three orders of magnitude smaller (as shown in Figs. S10, S11), indicating the importance of the boundary condition.

## Discussion

In this work, we found that the projected band structure of an ordinary hexagonal phononic crystal with hard-wall boundary conditions is essentially the same as that of a strip of a topological valley Hall phononic crystal carrying valley Chern numbers. In particular, there are boundary-condition-induced chiral bulk modes that are locked to the valleys and are counterparts to the valley-locked edge modes in topological valley Hall crystals. Such CABSs have similar robust transport characteristics as the edge modes in valley Hall phononic crystals, and their robustness is demonstrated both numerically and experimentally. Compared to the topological boundary states[8–18] obtained by using the bulk to induce robust boundary conduction, the CABSs are obtained using the boundary to control the bulk. In addition to experimentally demonstrating that CABSs can carry information around sharp bends and through structural defects, we also showed that boundary-condition modification can induce valley-selected transport, complete valley state conversion from K to K' and achieve valley focusing. Our work offers a new mechanism for controlling chiral valley transport, and this concept can be applied to other kinds of waves, such as EM waves.

## Methods
### Simulations
All the simulations in this work were carried out using the acoustics module of COMSOL Multiphysics. The speed of sound and the air density used are 349 m/s and 1.165 kg/m³, respectively. In Figs. 2b, 3a, b, 4a–c, 5a, the open boundary conditions were assumed at the entrance and exit ends, while the hard boundary conditions were assumed in the other directions.

### Experimental measurements
In the experiments, all the samples were fabricated by a 3D printing technology using resin. The sound signal are scanned by a movable microphone (diameter ~0.7 cm, B&K Type 4187) while fixing another identical microphone as a phase reference. The phase and amplitude of the pressure field can be obtained by a multiplex analyzer system (B&K Type 3560B).

## Data availability
The data that support the findings of this study are available from the corresponding authors upon reasonable request.

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

## Acknowledgements
We thank Prof. Z.Q. Zhang for fruitful discussions. This work is supported by the Research Grants Council of Hong Kong through grants 16310420, 16303119, AoE/P-502/20, the National Key R&D Program of China (Grant No. 2018YFA0305800), and the National Natural Science Foundation of China (Grants No. 11890701, No. 11774275, and No. 11974262). L.Z. was supported by National Natural Science Foundation of China Grant No. 12074230, National Key R&D Program of China under Grants No. 2017YFA0304203 and the Fund for Shanxi 1331 Project, and Shanxi Province 100-Plan Talent Program.

## Author contributions
M.W., Z.L., and C.T.C. conceived the idea. M.W. did the simulations and designed the experimental samples. M. W., R.-Y.Z., L.Z., Z.L., and C.T.C. did the theoretical analysis. M.W., Q.M., S.L., and M.K. carried out the measurements and did the experimental analysis. Z.L. and C.T.C. supervised the whole project. M.W., R.-Y.Z., Z.L., and C.T.C. wrote the draft.

## Competing interests
The authors declare no competing interests.
