## [Peer review file · Nature Communications]

REVIEWER COMMENTS

Reviewer #1 (Remarks to the Author):

In the manuscript "Observation of boundary induced chiral anomaly bulk states and their transport properties", the authors used a specific boundary condition to engineer the dispersions of the bulk states, and observed the bulk modes with so-called chiral anomaly inside the bandgap opened by finite crystal size. I think the novelty of this paper should be carefully justified for three reasons:

1. For crystals with finite sizes, all their eigenstates are naturally determined by both the bulk Hamiltonians and the boundary conditions, so it is not quite surprising that one can engineer the dispersions of some states by carefully tuning the boundary conditions.
2. The authors gave a simple theoretical explanation on their results by using a massive Dirac Hamiltonian with spatially varying effective mass $m(y)$. This is quite similar to their previous work [Ref. 17]. The only difference is that the effective mass $m(y)$ in the cladding region is finite in their previous work and assumed infinitely large in this work. This similarity makes the results in the current manuscript quite physically trivial.
3. The authors simply assumed that hard-wall boundary condition can be effectively regarded as the boundary between two regions with mass $m(y)$ equals 0 and $\pm\infty$. This assumption should be carefully justified.
 - 1). How could the authors determine the signs of the effective mass for $y > L$ and $y < 0$? Specifically, why did they assume $m(y > L) = +\infty$ and $m(y < 0) = -\infty$ rather than $m(y > L) = -\infty$ and $m(y < 0) = +\infty$?
 - 2). The current theoretical explanation cannot explain the more generic cases when the positions of the boundaries are slightly shifted and do not strictly terminate along the middle lines of the unit cells in the outermost layers of the samples (for instance, the boundaries conditions employed in the region 4 of Fig. 4b). It seems that the current theoretical explanation is simply a back derivation of the simulated results in Fig. 1c.

I strongly suggest the authors to upgrade their theoretical explanation to give a more generic and accurate mathematical description on the influence of the boundary conditions. For example, it should be possible to use the so-called boundary matrix method [Phys. Rev. B 77, 085423 (2008); Ref. 28] to analytically solve the wavefunctions and dispersions of all the states near the K and K' points.

Below are some technical comments on the current manuscript:

1. The authors should give more detailed description on the modified boundary at the joint region in Fig. 4c. What is the relationship between the reflection coefficient and the number of unit cells with modified boundary condition? Could the author give more simulated results?
2. In Fig. 6c, the authors conducted simulation at the optical wavelength. Could the authors clarify the polarization direction of the optical modes? Besides, could the authors provide the detailed geometric

configuration of the waveguide and the resonator, for example the relative positions between them. Could the authors provide the simulation transmission spectra of this simulation and clarify Fig. 6c belongs to which whispering gallery mode?

4. In Fig. 6d, the authors provide the numerical simulation with the resonator changed to a solid triangle, and claimed that the resonance disappeared. Such structure should also support many resonances with different resonant frequencies. Providing the field distribution at off-resonant frequencies in Fig. 6d cannot lead to the conclusion that “the resonance effect disappears”. The authors should clarify this point.

5. The demonstrated chiral bulk modes have the same modal profiles as the topological waveguide modes in the region B in Fig. 1a of [Ref. 17]. The modal profiles have nearly plane wavefront and do not exhibit net angular momentum. The authors should clarify this when they refer to the word “chiral” in the manuscript.

In conclusion, I think the authors should use a more accurate and generic theoretical model to clarify the underlying new physics of their works and justify their novelty beyond their previous work [Ref. 17]. Besides, the authors should provide more details about their simulation and experimental results.

Reviewer #2 (Remarks to the Author):

The manuscript experimentally demonstrates chiral anomaly bulk states imposing specific boundary conditions on an ordinary phononic crystal. The robustness of the chiral anomaly bulk states and the valley focusing are discussed and verified by experiments. In addition, the principle can be extended to the EM waves. In my opinion, the main idea of the chiral anomaly bulk states in this paper is interesting. However, some critical issues as listed below should be addressed.

1- In Fig. 3a, when the wave propagates through the 120-degree bend, the energy almost concentrates on one corner. Can the author explain the phenomenon?

2- In Fig. 3, the experimental Fourier spectra of the acoustic fields only possess the information when the wave hasn't passed through the bends or the disorders. Fourier spectra of the acoustic fields when the wave has passed through the perturbation should be added.

3- Based on the field patterns in Figs. 4(c), boundary modification causes nonnegligible reflection. However, the discussion and interpretation are not complete. The author should explain the mechanism of the boundary modification clearly.

4- In previous works, the robustness of edge states is confirmed by comparing the transmission in the structure with/without bends/disorders. In this manuscript, the current experiment results showing the momentum space can't validate the robustness of chiral anomaly bulk states.

REVIEWER COMMENTS

Reviewer #1 (Remarks to the Author):

In the manuscript “Observation of boundary induced chiral anomaly bulk states and their transport properties”, the authors used a specific boundary condition to engineer the dispersions of the bulk states, and observed the bulk modes with so-called chiral anomaly inside the bandgap opened by finite crystal size. I think the novelty of this paper should be carefully justified for three reasons:

1. For crystals with finite sizes, all their eigenstates are naturally determined by both the bulk Hamiltonians and the boundary conditions, so it is not quite surprising that one can engineer the dispersions of some states by carefully tuning the boundary conditions.

Reply: We thank the reviewer for carefully reviewing the paper and for the constructive comments. In the previous studies, although it is known that the dispersions of some states can be manipulated by carefully tuning the boundary conditions, it is usually impossible to know the exact dispersion only from a qualitative property of the boundary. In particular, a definite correspondence between chiral states and boundary condition is unknown. That is to say, it is uncertain what parameters can obtain the chiral state before numerically scanning the relevant parameters of the boundary. Compared with the previous works, the key theoretical innovation of our article is that we demonstrate the local symmetry of the boundary can deterministically determine the dispersion of certain bulk bands in acoustic and electromagnetic full-wave systems. We show that simply terminating the PC along the local mirror-invariant lines of the outermost unit cells of the sample encased by hard boundaries can guarantee the existence of the chiral bulk states by symmetry compatibility selection.

2. The authors gave a simple theoretical explanation on their results by using a massive Dirac Hamiltonian with spatially varying effective mass $m(y)$. This is quite similar to their previous work [Ref. 17]. The only difference is that the effective mass $m(y)$ in the cladding region is finite in their previous work and assumed infinitely large in this work. This similarity makes the results in the current manuscript quite physically trivial.

Reply: We thank the reviewers' comment. We would like to clarify that the present theory is not a simple use of the Jackiw-Rebbi model, i.e., the correspondence between the Dirac mass soliton and the existence of domain-wall states. On the contrary, our innovation is the discovery that the local symmetry selection mechanism can be mapped to the limiting case of the Jackiw-Rebbi model, hence making the existence of bulk chiral states more comprehensible.

Although our previous work [Ref. 17] and this work can get similar band structures, their underlying principles are different. In the previous work [Ref. 17], the structure is a direct application of the Jackiw-Rebbi model where the waveguide (domain B) is essentially a domain-wall of the two outside domains (A and C), and the existence of chiral waveguide states relies on the nontrivial valley topology of the domains A and C sandwiching domain B. However, for the waveguide structure in this paper, no domain possesses a nontrivial bulk topology, hence the chiral waveguide states are not the usual topological domain-wall/boundary modes but they are induced by the special boundary condition according to the local symmetry selection mechanism. In addition, our theory reveals the implicit connection between the two distinct mechanisms. Therefore, although the Jackiw-Rebbi model is well-known, uncovering the implicit correlation between the local symmetry selection mechanism and the Jackiw-Rebbi model is not trivial. Another important point to note is that the previous work is a variation of "valley" topology physics, which uses a bulk topological invariant to control transport. In this new paper, there are no bulk topological invariants, and the novel transport is enabled by boundary conditions. In short, the transport is controlled by bulk properties in prior valley topology work, while transport is controlled by boundary property in this work.

Despite having similarity in projected band dispersions, the different essential physical mechanisms make the chiral bulk modes (CBSs) in the two systems behave differently in some cases. For example: 1. For the ABC structure [Ref. 17], the CBSs can smoothly pass through the waveguides bent at 60-degree since the chirality of the valley-polarized CBSs is independent of the direction of the waveguide due to the uniform valley topology of the two domains sandwiching the two segments of waveguides. While for the structure in this work,

the CBSs cannot pass through the waveguide bent at 60-degree, since the chirality of the CBSs selected by the local symmetry of the boundary depends on the direction of the waveguide, making CBSs in the two segments of waveguides possess opposite chiralities. 2. It is easier to control the transport of CBSs in this paper than the ABC structure [Ref. 17]. As shown in Fig. 4a,b of the main text, by modifying the boundaries slightly, the CBSs can be controlled to pass or not pass through the 120-degree waveguide. Meanwhile, perfect valley state conversion can be achieved by slightly modifying the boundaries as shown in Fig. 4c. These phenomena are not easy to fulfill in the ABC structure [Ref. 17], since changing the transport of CBSs requires changing the bulk topology in domain A and C. Again, this relative ease of control of transport is because transport is controlled by boundary property in this work, while transport is controlled by bulk properties in prior valley topological systems.

3. The authors simply assumed that hard-wall boundary condition can be effectively regarded as the boundary between two regions with mass $m(y)$ equals 0 and $\pm\infty$. This assumption should be carefully justified.

1). How could the authors determine the signs of the effective mass for $y > L$ and $y < 0$? Specifically, why did they assume $m(y > L) = +\infty$ and $m(y < 0) = -\infty$ rather than $m(y > L) = -\infty$ and $m(y < 0) = +\infty$?

Reply: We thank the reviewers for raising this thoughtful question. Indeed, the correspondence between the sign of the Dirac mass and the local symmetry selection mechanism is one of the key discoveries of this work. As we expressed in the main text, for a

Dirac Hamiltonian with a step mass $m(y) = \begin{cases} m_1 & y < 0 \\ 0 & 0 \leq y \leq L \\ m_2 & y > L \end{cases}$, the bound state solution at

$k_x = 0$ and $\omega = 0$ (i.e. at the Dirac point) is given by

$$\psi = \begin{cases} e^{|m_1|y} \begin{pmatrix} 1 \\ \text{sgn}(m_1) \end{pmatrix} & y \leq 0 \\ \begin{pmatrix} 1 \\ \pm 1 \end{pmatrix} & 0 \leq y \leq L \\ e^{-|m_2|(y-L)} \begin{pmatrix} 1 \\ -\text{sgn}(m_2) \end{pmatrix} & y \geq L \end{cases}$$

The continuity of the wave function at $y = 0$ and L requires that this equation have only two

solutions: (1) $m_1 > 0$, $m_2 < 0$, and $\psi = \begin{pmatrix} 1 \\ 1 \end{pmatrix}$ (bulk even mode with respect to mirror- y operator σ_x (see main text)) when $0 \leq y \leq L$; (2) $m_1 < 0$, $m_2 > 0$, and $\psi = \begin{pmatrix} 1 \\ -1 \end{pmatrix}$ (bulk odd mode) when $0 \leq y \leq L$. This means that when the two $m(y) \neq 0$ domains ($y \leq 0$ and $y \geq L$) have opposite mass terms, **the sign of the Dirac masses $\text{sgn}(m_1) = -\text{sgn}(m_2)$ can select only one of the bulk modes with a certain mirror- y parity**, to be confined in the mid-domain.

On the other hand, since the real boundary of the acoustic waveguide is rigid, where the normal velocity component must be zero, only the mirror- y even bulk mode at the Dirac point is compatible with the rigid boundary condition. Comparing with the above effective Dirac Hamiltonian model, we found that the Dirac step mass with $m_1 = -m_2 \rightarrow +\infty$ rather than the opposite case with $m_1 = -m_2 \rightarrow -\infty$ can be mapped to the rigid acoustic boundary of the waveguide, since only the former one can support a mirror- y even bulk mode $\psi = \begin{pmatrix} 1 \\ 1 \end{pmatrix}$. This means the step Dirac mass with $m_1 = -m_2 \rightarrow +\infty$ is the sole solution (We are sorry that we carelessly posted a wrong mass distribution of Fig.1d was in the last version, and we have revised it in this revised manuscript). We have also added more discussions to explain this issue in the main text.

2). The current theoretical explanation cannot explain the more generic cases when the positions of the boundaries are slightly shifted and do not strictly terminate along the middle lines of the unit cells in the outermost layers of the samples (for instance, the boundaries conditions employed in the region 4 of Fig. 4b). It seems that the current theoretical explanation is simply a back derivation of the simulated results in Fig. 1c.

I strongly suggest the authors to upgrade their theoretical explanation to give a more generic and accurate mathematical description on the influence of the boundary conditions. For example, it should be possible to use the so-called boundary matrix method [Phys. Rev. B 77, 085423 (2008); Ref. 28] to analytically solve the wavefunctions and dispersions of all the states near the K and K' points.

Reply: We thank the referee for this useful suggestion. According to the referee's suggestion, here we use a boundary matrix method [see for example, Phys. Rev. B 77, 085423 (2008) and Ref. 28] to analytically solve the wavefunctions and dispersions of all the states near the K point (the case in K' point is similar, since the system has time-reversal symmetry).

The physics near the K valley is effectively characterized by the 2D Dirac Hamiltonian:

$$H = \sigma_x k_x v_D - i \partial_y \sigma_y v_D, \quad (\text{S1})$$

where k_x denotes the parallel wavevector relative to the K point. The bulk eigenstates at the K valleys satisfy the static 2D Dirac equation

$$H|\varphi(k_x, y)\rangle = \varepsilon|\varphi(k_x, y)\rangle. \quad (\text{S2})$$

Since the system has rigid boundaries at $y = 0$ and $y = L$, the wave function bounded by the rigid boundaries can be expressed as the superposition of the two linearly independent bulk eigenstates at each k_x :

$$|\varphi(k_x, y)\rangle = \alpha_1 |\varphi_1(k_x, y)\rangle + \alpha_2 |\varphi_2(k_x, y)\rangle \quad (0 \leq y \leq L), \quad (\text{S3})$$

where the two bulk eigenstates $|\varphi_1(k_x, y)\rangle = \begin{pmatrix} (k_x - u)v_D \\ \varepsilon \end{pmatrix} \exp(uy)$ and $|\varphi_2(k_x, y)\rangle = \begin{pmatrix} (k_x + u)v_D \\ \varepsilon \end{pmatrix} \exp(-u(y - L))$ with $u = \sqrt{k_x^2 - \varepsilon^2/v_D^2}$ are solved from Eq. (S2).

Hence, we can get

$$\begin{aligned} |\varphi(k_x, y)\rangle &= \alpha_1 \cdot \begin{pmatrix} (k_x - u)v_D \\ \varepsilon \end{pmatrix} \cdot \exp(uy) + \alpha_2 \cdot \begin{pmatrix} (k_x + u)v_D \\ \varepsilon \end{pmatrix} \cdot \exp(-u(y - L)) \\ &= \begin{pmatrix} (k_x - u)v_D \exp(uy) & (k_x + u)v_D \exp(-u(y - L)) \\ \varepsilon \exp(uy) & \varepsilon \exp(-u(y - L)) \end{pmatrix} \begin{pmatrix} \alpha_1 \\ \alpha_2 \end{pmatrix} \quad (0 \leq y \leq L) \end{aligned} \quad (\text{S4})$$

Next we will examine the effect of bulk state when changing boundaries. The effect of the boundary tuning at $y = 0$ and $y = L$ can be represented as

$$\widehat{M}_1 |\varphi(k_x, 0)\rangle = |\varphi(k_x, 0)\rangle, \quad (\text{S5})$$

$$\widehat{M}_2 |\varphi(k_x, L)\rangle = |\varphi(k_x, L)\rangle. \quad (\text{S6})$$

Here \widehat{M}_1 and \widehat{M}_2 are 2x2 unitary matrix satisfying $\widehat{M}^2 = 1$. Two rigid boundaries makes that the bulk states can only propagate along the x -direction, while the y -direction is forbidden, hence $\{\widehat{M}, \sigma_y\} = 0$ [Phys. Rev. B 77, 085423 (2008)].

By virtue of these constraints, the two the boundary matrixes \widehat{M}_1 and \widehat{M}_2 can be expressed as

$$\widehat{M}_1(\theta_1) = \sigma_x \sin\theta_1 + \sigma_z \cos\theta_1, \quad (\text{S7})$$

$$\widehat{M}_2(\theta_2) = \sigma_x \sin\theta_2 + \sigma_z \cos\theta_2, \quad (\text{S8})$$

where θ_1 and θ_2 are determined by the lower (upper) boundary condition of the real system. The wavefunction $|\varphi(k_x, y)\rangle$ at $y = 0$ and $y = L$ should satisfy Eqs. (S5, S7) and Eqs. (S6, S8) respectively, which indicates $|\varphi(k_x, 0)\rangle$ and $|\varphi(k_x, L)\rangle$ are the eigenstates of \widehat{M}_1 and \widehat{M}_2 corresponding to the eigenvalue of $+1$, respectively:

$$|\varphi(k_x, 0)\rangle = \beta_1 \begin{pmatrix} \sin\theta_1 \\ 1 - \cos\theta_1 \end{pmatrix}, \quad (\text{S9})$$

$$|\varphi(k_x, L)\rangle = \beta_2 \begin{pmatrix} \sin\theta_2 \\ 1 - \cos\theta_2 \end{pmatrix}. \quad (\text{S10})$$

There are four unknown parameters $\alpha_1, \alpha_2, \beta_1, \beta_2$ for four linear Eqs. (S4, S9, S10).

$$R \begin{pmatrix} \alpha_1 \\ \alpha_2 \\ \beta_1 \\ \beta_2 \end{pmatrix} = \begin{pmatrix} (k_x - u)v_D & (k_x + u)v_D e^{uL} & -\sin\theta_1 & 0 \\ \varepsilon & \varepsilon e^{uL} & -1 + \cos\theta_1 & 0 \\ (k_x - u)v_D e^{uL} & (k_x + u)v_D & 0 & -\sin\theta_2 \\ \varepsilon e^{uL} & \varepsilon & 0 & -1 + \cos\theta_2 \end{pmatrix} \begin{pmatrix} \alpha_1 \\ \alpha_2 \\ \beta_1 \\ \beta_2 \end{pmatrix} = \begin{pmatrix} 0 \\ 0 \\ 0 \\ 0 \end{pmatrix} \quad (\text{S11})$$

This equation has nonzero solutions when the determinant of the matrix R vanishes:

$$\det R(\varepsilon, k_x, \theta_1, \theta_2) = 0. \quad (\text{S12})$$

For a fixed boundary condition, θ_1 and θ_2 are determined, and only ε and k_x is unknown in Eqs. (S11) (This equation is hard to get analytical solution, but their relationship is deterministic). Hence, the relationship between ε and k_x can be obtained:

$$\varepsilon = f(k_x, \theta_1, \theta_2). \quad (\text{S13})$$

Especially, when $\theta_1 = \theta_2 = \frac{\pi}{2}$, we have $\widehat{M}_1 = \widehat{M}_2 = \sigma_x$ is coincident with the mirror- y operator of the Dirac Hamiltonian (see main text) and $|\varphi(k_x, 0)\rangle = |\varphi(k_x, L)\rangle \propto \begin{pmatrix} 1 \\ 1 \end{pmatrix}$ indicates the boundary conditions select the bulk even modes with $\varepsilon = k_x v_D$, which corresponds to the band structure in Fig. 1c. The existence of boundaries makes additional boundary potential to be applied in the vicinity of the boundaries. We take the additional boundary potential $V_1(y)$ applied in the vicinity of the bottom edge ($V_1(0 < y < y_0) \neq 0$ and $V_1(y \geq y_0) = 0$ with $y_0 \rightarrow 0$) as an example.

The modified Hamiltonian is

$$H = \sigma_x k_x v_D - i \partial_y \sigma_y v_D + V_1(y). \quad (\text{S14})$$

The transfer matrixs $T(y_1, y_2)$ are introduced to connect the wavefunctions $|\varphi_k(y)\rangle$ from y_1 to y_2 .

$$|\varphi(k_x, y_1)\rangle = T(y_1, y_2)|\varphi(k_x, y_2)\rangle, \quad 0 < y_2 < y_1 < L, \quad (\text{S15})$$

According to Eqs. (S2, S14, S15), the transfer matrix $T(y_1, y_2)$ can be obtained:

$$T(y_1, y_2) = \text{Pexp} \int_{y_2}^{y_1} \frac{1}{v_D} (\sigma_z k_x - iV_1(y)\sigma_y) \cdot dy, \quad (\text{S16})$$

where Pexp is a path-ordered exponential.

If no boundary potential is applied, the transfer matrix is T_0 . When $y \geq y_0$, the transfer matrix can be expressed as

$$T(y, 0) = T(y, y_0)T(y_0, 0) = T_0(y, y_0)T(y_0, 0), \quad (y \geq y_0), \quad (\text{S17})$$

and the bulk states can be written as

$$|\varphi(k_x, y \geq y_0)\rangle = T(y, y_0)|\varphi(k_x, y_0)\rangle. \quad (\text{S18})$$

According to Eqs. (S5, S18), the following relationship is established

$$[T(y, 0)\widehat{M}_1 T^{-1}(y, 0)]|\varphi(k_x, y_0)\rangle = |\varphi(k_x, y_0)\rangle. \quad (\text{S19})$$

\widehat{M}_{1eff} is the equivalent boundary matrix and satisfy

$$\widehat{M}_{1eff}|\varphi(k_x, y_0)\rangle = |\varphi(k_x, y_0)\rangle. \quad (\text{S20})$$

Comparing Eqs. S19 and S20, we can get

$$\widehat{M}_{1eff} = T(y, y_0) \cdot \widehat{M}_1(\theta_1) \cdot T^{-1}(y, y_0). \quad (\text{S21})$$

Since y_0 is very small compared with wavelength, the Eq. (S16) can be simplified to

$$T(y_0, 0) = \exp(i\theta_{V_1}\sigma_y), \quad \theta_{V_1} = -\int_0^{y_0} \frac{V_1(y)}{v_D} dy. \quad (\text{S22})$$

Combining Eqs. (S21) and (S22), one obtains a simple form of the effective boundary matrix which is identical to the original boundary matrix up to a parameter shift θ_{V_1} :

$$\widehat{M}_{1eff} = \widehat{M}_1(\theta_1 + \theta_{V_1}). \quad (\text{S23})$$

The additional boundary potential $V_2(y)$ applied in the vicinity of the upper edge ($y \rightarrow L$) has the similar effect:

$$\widehat{M}_{2eff} = \widehat{M}_2(\theta_2 + \theta_{V_2}). \quad (\text{S24})$$

These verify that modifying the boundary matrix can simulate the effect of adding additional boundaries. Hence when two additional boundary potentials $V_1(y)$ and $V_2(y)$ are added, the energy ε will the following change:

$$\varepsilon = f(k_x, \theta_1 + \theta_{V_1}, \theta_2 + \theta_{V_2}). \quad (\text{S25})$$

Moreover, we show the influence of the boundary truncation positions on the band structure (simulated by COMSOL) in Fig. S4. In the process of changing the boundary truncation

positions at two outmost layers, we ensure that the PC waveguide maintains mirror-y symmetry. Fig. S2a shows the band structure of the PC slab with a width varying from 10 to 12 layers (normalized by $\sqrt{3}/2a$) by shifting the truncation positions. We focus on the frequencies near 15.5 kHz, which are in the frequency range of the CABSs. When the width of the PC slab increases from 10 to 12 layers, the band structure changes continuously, and no band gap appears. Compared to the CABSs for the PC with 10 layers, the slope of the CABS is reversed at each valley when the layer number increases to 11, while the dispersion is restored when the layer number is 12. These results show that shifting the truncation position one lattice constant at each boundary represents a complete change cycle for the CABSs, therefore, the band structure can be adjusted conveniently by adjusting the boundary truncation positions. The change in the group velocity of the CABS as a function of the PC width at approximately 15.5 kHz is shown in Fig. S2b. Compared to the group velocity when the layer number is 10, the CABS group velocity is reversed when the layer number is 11, while it is restored when the layer number is 12. These results are consistent with the change in the band structure.

We have added the theoretical explanation and simulation results on the influence of the boundary conditions in the Supplementary Information.

Fig. S2. a, The change in band dispersion of the PC with increasing slab layers. **b**, The change in the group velocities of the CABS near 15.5 kHz as a function of the PC width.

Below are some technical comments on the current manuscript:

1. The authors should give more detailed description on the modified boundary at the joint region in Fig. 4c. What is the relationship between the reflection coefficient and the number of unit cells with modified boundary condition? Could the author give more simulated results?

Reply: We thank the referee for the questions. We have calculated the transmission coefficients with five different boundary conditions, namely the unmodified boundary and

four truncated boundaries (remarked by the four colored lines in Fig. S6a, and the angle of the truncation boundary to the x direction is kept at 30 degree), as shown in Fig. S6b. The transmission in the shaded area is close to unity in the case of truncated boundary 2 (colored in green, i.e. the same modified boundary condition with Fig. 4c). The oscillations in the transmittance are due to Fabry-Perot effect, as the total length of the whole waveguide is finite.

Fig. S6. **a**, Diagrams showing four different truncated boundaries at the joint region for the 60-degree waveguide. **b**, From top to bottom are the transmissions of unmodified waveguide and four waveguides with truncated boundaries 1-4.

2. In Fig. 6c, the authors conducted simulation at the optical wavelength. Could the authors clarify the polarization direction of the optical modes? Besides, could the authors provide the detailed geometric configuration of the waveguide and the resonator, for example the relative positions between them. Could the authors provide the simulation transmission spectra of this simulation and clarify Fig. 6c belongs to which whispering gallery mode?

Reply: We thank the referee for the questions. The electric polarization direction of the optical modes is along the z direction, so that the modes are the transverse magnetic (TM) modes.

The widths of waveguide and the resonator are all $4\sqrt{3}a$, and the spacing between the waveguide and the resonator is $\frac{\sqrt{3}}{2}a$, as shown in Fig. S9. The simulated transmission spectrum of the waveguide is shown in Fig. S10a and the calculated eigenmode Q factor ($\lg(Q)$) for the resonator is shown in Fig. S10b. Two almost degenerate whispering gallery modes at 482 THz are shown in the right panel of Fig. S10, which correspond to the excited field inside the triangular resonator shown in Fig. 6c of the main text. We have added these descriptions and pictures in main text and supplementary information.

Fig. S9. Diagram of the waveguide and the resonator.

Fig. S10. **a**, The simulation transmission spectra of the waveguide in Fig. 6c. The dielectric constant in the triangular resonator is $12 + 0.002 * i$. The straight waveguide has no gain incorporated. The small gain in the triangle resonator results in amplified transmission at some resonant frequencies. **b**, Calculated eigenmode Q factor ($\lg(Q)$) for the resonator in Fig. 6c. Right panel: the two eigenmode at 482 THz.

4. In Fig. 6d, the authors provide the numerical simulation with the resonator changed to a solid triangle, and claimed that the resonance disappeared. Such structure should also support many resonances with different resonant frequencies. Providing the field distribution at off-resonant frequencies in Fig. 6d cannot lead to the conclusion that “the resonance effect disappears”. The authors should clarify this point.

Reply: We thank the referee’s question. When the resonator changed to a solid triangle (Fig. 6d), the Q factor ($\lg(Q)$) for this resonator, as shown in Fig. S11, is about three orders of magnitude smaller than the resonator in Fig. 6c. To be more accurate, we changed the previous sentence “the resonance effect disappears” to “the resonance effect has become significantly weaker, and the Q factor is about three orders of magnitude smaller”.

Fig. S11. **a**, Calculated eigenmode Q factor ($\lg(Q)$) for the solid triangle in Fig. 6d.

5. The demonstrated chiral bulk modes have the same modal profiles as the topological waveguide modes in the region B in Fig. 1a of [Ref. 17]. The modal profiles have nearly plane wavefront and do not exhibit net angular momentum. The authors should clarify this when they refer to the word “chiral” in the manuscript.

Reply: We thank the referee’s comment. In this paper, the “chiral” means that the transportation in only one direction, instead of two, is permitted in each single valley (K or K’). We are using such terminology in accordance with the usage in some recently published papers. For example, Ref. 28 (“Observation of chiral edge states in gapped nanomechanical graphene”) has already used the word “chiral” to describe this, and we are consistent with it.

In conclusion, I think the authors should use a more accurate and generic theoretical model to clarify the underlying new physics of their works and justify their novelty beyond their previous work [Ref. 17]. Besides, the authors should provide more details about their simulation and experimental results.

Reply: We thanks for the referees’ suggestion. Now we have made a more comprehensive and generic theoretical model to clarify the underlying new physics of our works. As mentioned earlier, we have explained in detail the key differences between this paper and the previous paper in terms of physical principles and application effects. Meanwhile more details about their simulation and experimental results are added.

Reviewer #2 (Remarks to the Author):

The manuscript experimentally demonstrates chiral anomaly bulk states imposing specific boundary conditions on an ordinary phononic crystal. The robustness of the chiral anomaly bulk states and the valley focusing are discussed and verified by experiments. In addition, the principle can be extended to the EM waves. In my opinion, the main idea of the chiral anomaly bulk states in this paper is interesting. However, some critical issues as listed below should be addressed.

Reply: We thank the referee for approving the main idea of our work. We have carefully considered all the questions/suggestions, and responded point by point.

1- In Fig. 3a, when the wave propagates through the 120-degree bend, the energy almost concentrates on one corner. Can the author explain the phenomenon?

Reply: Indeed, the field distribution around the corner is determined by the details of the of the corner morphology. There is no known a priori method to predict the field distribution in that coupling region. Nevertheless, here we will try to give an intuitive understanding of this phenomenon. Let us add two auxiliary lines to partition the 120-degree waveguide, as shown in Fig. R1, into three regions: domains 1 and 2 are two equivalent waveguides and domain 3 is the coupling region. The EM wave needs to go through the coupling region from domain 1 to domain 2. We expect that the wave would prefer to select to concentrate in the region which offers the best impedance matching for wave coupling, and the coupling region is “big enough” for such a region to exist. From the numerical results, we note that most waves pass through the coupling region at the top right corner of domain 3. We may alternatively give an a posteriori rationalization that the wave prefers to take the shortest path around the corner, and the upper region has a higher concentration of wave amplitude due to the closer separation between the two waveguides at this location.

Fig. R1. The two black dashed lines divide the 120-degree waveguide into three regions, where domain 3 is the coupling region.

2- In Fig. 3, the experimental Fourier spectra of the acoustic fields only possess the information when the wave hasn't passed through the bends or the disorders. Fourier spectra of the acoustic fields when the wave has passed through the perturbation should be added.

Reply: We thank the referee's suggestion. We have performed new experimental measurements, and the Fourier spectra of the acoustic fields when the wave has passed through the perturbation have been added in Fig. 3, as shown below. We also have updated the Fig. 3 in the main text.

Fig. 3. a,b, Simulated acoustic pressure field distributions at 15.5 kHz for a point source (red star) on the left of the 120-degree-bend PC and the disordered PC (the cylinder diameters of the 32 units are randomly distributed between 5-9 mm in the black rectangle). Inset picture: the corresponding experimental Fourier spectra of the acoustic field in the magenta dashed rectangle in **a** and **b**.

3- Based on the field patterns in Figs. 4(c), boundary modification causes nonnegligible reflection. However, the discussion and interpretation are not complete. The author should explain the mechanism of the boundary modification clearly.

Reply: We thank the referee's question. The 60-bend waveguide, as shown in Fig. 4a, can be seen as consisting of two waveguides in domain 1 and 3, and the junction of the two waveguides is the coupling region. The sound wave cannot pass through the 60-bend waveguide smoothly. In order to change this state from one valley to another, we can make some modifications in the coupling region. We note that the two waveguides (domain1 and 3) are symmetric about the middle line of the coupler (i.e. the line connecting the two corners). To maintain the symmetry, we truncate the coupling region along an angle of 30 degrees from

the x -direction. At the same time, we know that the transmittance of the waveguide is related to its width. Therefore, under the premise of ensuring symmetry, we cut different widths in coupling region to see which one has the highest transmission, as shown in Fig. S6. And we find the truncated boundary 2 has the best results, the transmission of the 60-bend waveguide (green color in Fig. S6b) closes to 1 in the shaded frequency range. As a result, the sound wave can pass through the 60 degree-bend in Fig. 4c, and the reflection is highly suppressed (which is proved by the Fourier transformation in domain 1 where only K valley states are excited), after the boundary modification. We have added these discussions in Supplementary Information.

Fig. S6. **a**, Diagrams showing four different truncated boundaries at the joint region for the 60-degree waveguide. **b**, From top to bottom are the transmissions of unmodified waveguide and four waveguides with truncated boundaries 1-4.

4- In previous works, the robustness of edge states is confirmed by comparing the transmission in the structure with/without bends/disorders. In this manuscript, the current experiment results showing the momentum space can't validate the robustness of chiral anomaly bulk states.

Reply: We thank for the referee's question. The band dispersion in Fig. 1c show that only two channels (one is in the K valley while the other is in the K' valley) in the waveguide allow energy transfer. The current experimental results in Fig. 3a and Fig. 3b show that in the waveguide with bends (Fig. 3a) and disorders (Fig. 3b), only the K valley state is well excited, while the K' valley state is strongly inhibited. The single valley excitation in the upstream of the bends and disorders means that the transmission of sound waves in waveguide is not disturbed by the bending defect, which can validate the robustness of chiral anomaly bulk states.

Moreover, we theoretically compared the transmission in the straight waveguide, the 120-bend waveguide, and the waveguide with disorders at the frequency of CABSs, as shown in Fig. S4. The results show that the transmissions of all three cases are similar in the shadow frequency range, which again verifies the robustness of chiral bulk states. We have added the relevant simulation results in the Supplementary Information.

Fig. S4. The transmissions of the straight waveguide without defects, 120-degree-bend waveguide and the disordered waveguide at the frequency of CABSs.

REVIEWERS' COMMENTS

Reviewer #1 (Remarks to the Author):

In the revised manuscript, the authors have added many solid theoretical and numerical results to successfully answer all my comments and questions. I think the revised manuscript is good and interesting for publication.

Reviewer #2 (Remarks to the Author):

The comments of this manuscript have been carefully addressed. I think the manuscript should be considered for publication in Nature Communications.